# Correlation Between Composition and Electrodynamics Properties in Nanocomposites Based on Hard/Soft Ferrimagnetics with Strong Exchange Coupling

**DOI:** 10.3390/nano9020202

**Published:** 2019-02-04

**Authors:** Munirah Abdullah Almessiere, Alex V. Trukhanov, Yassine Slimani, K.Y. You, Sergei V. Trukhanov, Ekaterina L. Trukhanova, F. Esa, A. Sadaqat, K. Chaudhary, Maxim Zdorovets, Abdulhadi Baykal

**Affiliations:** 1Department of Physics, College of Science, Institute for Research & Medical Consultations (IRMC), Imam Abdulrahman Bin Faisal University, P.O. Box 1982, 31441 Dammam, Saudi Arabia; malmessiere@iau.edu.sa; 2Department of Nano-Medicine Research, Institute for Research & Medical Consultations (IRMC), Imam Abdulrahman Bin Faisal University, P.O. Box 1982, 31441 Dammam, Saudi Arabia; abaykal@iau.edu.sa; 3SSPA “Scientific and practical materials research center of NAS of Belarus”, 220072 Minsk, Belarus; sv_truhanov@mail.ru (S.V.T.), kastor1986@yandex.ru (E.L.T.); 4South Ural State University, 454080 Chelyabinsk, Russia; 5National University of Science and Technology MISiS, 119049 Moscow, Russia; 6Department of Physics Research, Institute for Research & Medical Consultations (IRMC), Imam Abdulrahman Bin Faisal University, P.O. Box 1982, 31441 Dammam, Saudi Arabia; yaslimani@iau.edu.sa; 7School of Electrical Engineering, Faculty of Engineering, Universiti Teknologi Malaysia, Skudai-Johor 81310, Malaysia; kyyou@fke.utm.my; 8Physics and Chemistry Department, Faculty of Applied Sciences and Technology, Universiti Tun Hussein Onn Malaysia, Pagoh-Johor 81310, Malaysia; fahmir@uthm.edu.my; 9Mechanical and Energy Engineering Department, College of Engineering, Imam Abdulrahman Bin Faisal University, P.O. Box 1982, 31441 Dammam, Saudi Arabia; truhanov86@gmail.com; 10Physics Department, Faculty of Science, Univerity Teknology Malaysia, Johor Bahru-Johor 81310, Malaysia; kashif@utm.my; 11L.N. Gumilyov Eurasian National University, Astana 10008, Kazakhstan; m_zdorovets@gmail.com; 12The Institute of Nuclear Physics of Republic of Kazakhstan, Astana 10008, Kazakhstan; 13Ural Federal University named after the First President of Russia B.N. Yeltsin, Yekaterinburg 620002, Russia

**Keywords:** microwave absorption, nanosized composites, hard-soft ferrites, electromagnetic properties, reflection losses

## Abstract

In this work, Sr_0.3_Ba_0.4_Pb_0.3_Fe_12_O_19_/(CuFe_2_O_4_)_x_ (x = 2, 3, 4, and 5) as strongly exchange-coupled nanosized ferrites were fabricated using a one-pot sol–gel combustion method (citrate sol-gel method). The X-ray diffraction (XRD) powder patterns of the products confirmed the occurrence of pure, exchange-coupled ferrites. Frequency dependencies of the microwave characteristics (MW) were investigated using a co-axial method. The non-linear behavior of the MW with the composition transformation may be due to different degrees of Fe ion oxidation on the spinel/hexaferrite grain boundaries and strong exchange coupling during the hard and soft phases.

## 1. Introduction

Strongly correlated transition metal oxides exhibit a wide spectrum of unusual electronic and magnetic phenomena [1,2,3] caused by the cooperative effects of charge and spin ordering. This class of materials demonstrates such quantum phenomena as high-temperature superconductivity [4], Bose-Einstein condensation of magnons [5], and multiferroicity (the coexistence of magnetic and ferroelectric ordering) [6]. Functional materials with coexisting hard magnetic and soft magnetic properties at room temperature have attracted much attention [7,8]. The most interesting classes are the multiferroic and electromagnetic composites [9]. The coexistence of two separate magnetic phases may provide strong coupling between them and as a result lead to an improvement of the functional properties [10,11]. For example, it can lead to a modification of the initial electrical and magnetic properties as compared to the pure materials [12]. The main aim is to determine the correlation between chemical compositions (concentration ratio of different phases) and functional properties in composites. The maximal effect in composites can be reached by strong exchange coupling (magnetostatic coupling, intergranular interactions, microstructure effect).

Many researchers have been focused on complex metal oxides based on iron ions (hexaferrites, spinels, perovskites etc.). Barium M-type hexaferrite (BaFe_12_O_19_) and solid solutions based on it are the most attractive objects for investigation. These compounds have a magnetoplumbit structure—space group P6_3_/mmc (No. 194) with cell parameters a = b ≈ 5.90 Å, c ≈ 23.30 Å. BaFe_12_O_19_ is an important material for microwave applications due to its high saturation magnetization, low electrical conductivity and large magneto–crystalline anisotropy [13]. There are two main mechanisms for microwave absorption in BaFe_12_O_19_: 1. Domain boundary resonance; 2. Natural ferromagnetic resonance. Weakening of the transmitted electromagnetic radiation opens up a wide range of perspectives for microwave absorbers [14,15,16,17]. Hard and soft ferrites are technologically important materials owing to their special applications in data-storage media, microwave devices [18] and permanent magnets [19]. Because strong exchange-coupling occurs [20] between two soft and hard magnetic phases, an intensification of the microwave absorption can be observed [21,22,23,24,25]. Shen et al. [26] confirmed the advantages of exchange coupling ferrites.

In this study, Sr_0.3_Ba_0.4_Pb_0.3_Fe_12_O_19_/(CuFe_2_O_4_)_x_ (x = 2, 3, 4, and 5) composites were produced using a citrate sol-gel method. The correlation between the composition and microwave properties of the composites are discussed.

## 2. Materials and Methods

Sr_0.3_Ba_0.4_Pb_0.3_Fe_12_O_19_/(CuFe_2_O_4_)_x_ (x = 2, 3, 4, and 5) composites were produced usinga citrate sol-gel method [27,28,29,30]. Nitrates of corresponding ions (Fe^3+^, Ba^2+^, Pb^2+^, Cu^2+^) were mixed to a stoichiometric ratio by adding the citric acid and de-ionized water at a temperature of 355 K. The weight ratio between the citric acid and ion nitrites was 1.5:1. Following that, the mixture was slowly cooled to 298 K. A pH correction using citric acid and a chelation was done. This lead to a 3D-structure formation (nitrate-citrate xerogel). Then the xerogel was heated for the “dark gel” phase formation (water evaporation transforms xerogel in the next solution stage—“dark gel” with high viscosity). The “dark gel” was then heated to 523 K by self-ignition. It was accompanied by the formation of a large volume of gas. This first stage allowed us to obtain the initial powders. After pre-firing (at 733 K), samples were calcined at 1373 K for 2 h. The features of the crystal structure and phase compositions were investigated using X-ray diffraction (XRD) under Cu-Kα radiation (Rigaku D/MAX-2400, Japan). The peculiarities of the chemical composition and microstructure were analyzed using Scanning Electron Microscopy (SEM) (Hitachi S-4800, Japan) with Energy-dispersive X-ray spectroscopy (EDX). In addition, we used a high-resolution transmission electron microscopy (HRTEM) (FEI Titan S/TEM microscope, Netherlands). The frequency dependences of the permeability and permittivity were investigated using a co-axial method with an Agilent network analyzer in 8–12 GHz and 100–1000 MHz frequency ranges. The impedance of the co-axial line was normalized (Z = 50 Ohm).

Based on the obtained values for the real and imaginary parts of the permittivity and permeability (ε′, εʺ, μ′ and μʺ), the reflection coefficients were calculated by referring to the theory of propagation of an electromagnetic wave in the transmission line:(1)R˙=Z˙M−Z˙Z˙M+Z˙
where, Z˙M is the impedance of the composite (investigated) material and Z˙ is the impedance of the coaxial line (in this case 50 Ohm).

In general, the coaxial line impedance is determined by following equation:(2)Z˙=60ln(Dd)μ˙ε˙
where, D is the outer diameter of the coaxial cable, d is the inner diameter of the coaxial cable, μ is the complex permeability, and ε is the complex permittivity.

To calculate reflection losses, the following formulae were used:(3)R˙=μ˙ε˙−1μ˙ε˙+1
in dB
(4)|R˙|=20lg(μ˙ε˙−1μ˙ε˙+1)
where the modulus of the reflection coefficient shows the ratio of the amplitude of the reflected wave relative to the incident amplitude in dB.

## 3. Results and Discussion

### 3.1. Crystal Structure and Microstructure

The XRD patterns of the Sr_0.3_Ba_0.4_Pb_0.3_Fe_12_O_19_/(CuFe_2_O_4_)_x_ (x = 2, 3, 4, and 5) samples are presented in Figure 1.

The analysis of the XRD data proved the coexistence of two main phases in the samples: CuFe2O4 (JCPDS 34-0425) and BaFe_12_O_19_ (JCPDS 00-043-0002). The XRD data was processed using Rietveld Refinement (FullProf. Software). The soft magnetic phase (CuFe_2_O_4_) corresponds to the spinel structure with Space Group Fd-3m (No. 227)

The hard magnetic phase (Sr_0.3_Ba_0.4_Pb_0.3_Fe_12_O_19_) corresponds to the magneto-plumbite structure with Space Group P6_3_/mmc (No. 194). The analysis of the following parameters; Rwp (weighted profile R-value), Rexp (expected R-value), RB (Bragg R-factor), Rmag (magnetic R-factor) and χ2 (goodness-of-fit quality factor) was performed after refinement suggested that the investigated samples were of sufficiently good quality and the refinements are effective. The features of the crystal structure for each phase are shown in Table 1.

The values of a and c for the hard magnetic phase were slightly varied from 5.880 Å to 5.888 Å and from 23.104 Å to 23.128 Å respectively. The value for the soft magnetic phase was slightly varied from 8.323 Å to 8.411 Å. Negative deviations in unit cell parameters in comparison with bulk ceramics can be explained by the effect of surface compression upon the transition of crystallites to the nano level. For all samples, a structural phase transition was not detected. A phase transition for Cu-spinel from the cubic phase (SG Fd-3m) to the tetragonal phase (sp. gr. I41/amd) was no observed. These results demonstrate to us that the sol-gel method allowed for the growth of both hard and soft ferrites as strongly exchange-coupled composites [18]. This means that there are two separate phases (soft and hard magnetic phases) without chemical interactions with strong magnetostatic coupling (bias exchange between grains) that influences the electromagnetic properties of each phases.

The FE-SEM images and EDX spectra of S_r0.3_Ba_0.4_Pb_0.3_Fe_12_O_19_/(CuFe_2_O_4_)_x_ composite ceramic samples (x = 2, 3, 4, and 5) are presented in Figure 2. The chemical fractions of all elements confirmed the formation of the desired compositions in different samples. The HRTEM images (Figure 3) demonstrate values for distances between the atomic planes of 0.47 and 0.50 nm. This corresponds to the (102) and (100) atomic planes of hexaferrite. Distances between atomic planes of 0.20 nm is typical for the (400) atomic planes of spinels [17,29,30].

### 3.2. Electrodynamic Properties

The tangent of the dielectric loss angle (tgδ), dielectric permittivity (real ε′ and imaginary εʺ parts), magnetic permeability (real μ′ and imaginary μʺ parts) and impedance (Z) as functions of frequency were extensively measured by widely adopted coaxial [31,32] and waveguides methods [33,34,35]. Due to issues related to the measurement dynamical range, the waveguide method is appropriate for narrow frequency ranges only and limited by the size of the sample. Thus, to measure the electrodynamic parameters of the sample in a wide range, it is required to have different measuring sections of the waveguide and to prepare samples of a suitable size.

It is better to use a coaxial or long-line method to measure the electrodynamic parameters of the powder sample in wide-ranging frequencies [36].

Figure 4 and Figure 5 demonstrate frequency dependencies of the permittivity and electrical conductivity. Investigations were carried out at T = 300 K in the range of 8–12.5 GHz. Compacted Sr_0.3_Ba_0.4_Pb_0.3_Fe_12_O_19_/(CuFe_2_O_4_)_x_ composites (x = 2, 3, 4, and 5) were placed in the co-axial line (Z = 50 Ohm). Figure 4 demonstrates that the real (top) and imaginary (bottom) parts of the permittivity depend slightly on frequency.

In the investigated range, no critical changes in permittivity due to the absence of any electrical losses in composites were observed over 8–12.5 GHz. Some blurred peaks on curves (deviation from linearity) can be explained by some insufficient resonance due to geometric factors (interference losses). For the real part, there was an observed dispersion in values for samples with x = 2, 4. This non-linear behavior (decrease in permittivity only for samples with a determined concentration without any concentration dependence) may be the result of different degrees of oxidation on the spinel/hexaferrite grain boundaries with a high value of activation energy of conductivity.

Values for the imaginary part of permittivity and active electric conductivity were calculated using the following equations:(5)εʺ= ε′ × tgδ
(6)σa= εʺ× ε0 ω
where, tgδ is the loss tangent, σ a is the active part of electric conductivity, ε0 is dielectric constant, and ω, is the circular frequency (2πf).

Figure 5 demonstrates the frequency dependence of the electrical conductivity. The values measured for the conductivity are typical for highly doped semiconductors or composites where realized hopping conduction mechanism occur. The revealed blurred peaks are in agreement with the measurement of permittivity in the same MW range. Differences in conductivity values between compounds with x = 2 and 4; x = 3 and 5 were determined by the difference in the activation energy of the intergranular potential barrier.

Figure 6 demonstrates the frequency dependence of permeability (real and imaginary parts). Further investigations were carried out at T =3 00 K in the range 8–12.5 GHz.

There were no critical changes in permeability due to the absence of any magnetic losses in composites observed in this region. Chemical substitution in hexaferrites must shift the peak of the natural ferromagnetic resonance to the lower frequency region. The calculation of the magnetic permeability and dielectric permittivity of the S_r0.3_Ba_0.4_Pb_0.3_Fe_12_O_19_/(CuFe_2_O_4_)_x_ composites (x = 2, 3, 4, and 5) was performed using the Nicholson–Ross–Weer method (NRW) [37] by measuring the so called ‘scattering’ parameters of the coaxial line segment S11 and S21. Figure 7a and b demonstrate the frequency dependence of the S11–S21 parameters. These parameters correspond to the transition and reflection losses (attenuation in incident power of the radiation).

Figure 8 demonstrates the reflection losses (RL) for Sr0.3Ba0.4Pb0.3Fe12O19/(CuFe2O4)x composites (x = 2, 3, 4, and 5). The maximal values of RL for x = 3 and 5 are less than –4.5 dB thus the main losses are due to reflection rather than absorption. Using transmission (TL−transmission losses) and reflection (RL−reflection losses) coefficients, the absorption coefficient kabs was calculated as follows [22]:(7)kabs=10log (1−100.1ktr−100.1kref)

Figure 9 demonstrates the frequency dependence of absorption losses for the composites investigated in the frequency range 8–12 GHz. It is clear that no sufficient weakening was caused by electromagnetic absorption in this frequency range. It must be mentioned that this frequency range is a typical region of absorption for ferrites with a hexagonal structure. It means that the mixing of hard and soft magnetic phases in composites leads to a significant decrease in the microwave properties of hexagonal ferrites (hard magnets).

Figure 10 demonstrates the frequency dependence of absorption losses measured using a coaxial method in the frequency range 1 MHz–1 GHz. This frequency region was chosen due to the possibility of observing resonance reflection (NFMR) in the soft magnetic phase (spinel). It is noted that the presence of resonance (significant weakening of the reflected radiation) in this frequency range is typical of spinels [38,39]. Such behavior is due to a strong coupling between soft magnetic and hard magnetic phases that results in a weakening of the absorption for hard the magnetic phase and an increased reflection for soft magnetic phase as a function of the phase ratio concentration.

Figure 11 demonstrates the concentration dependence of the maximum value of reflection or resonant amplitude (A_res_) and of the resonant frequency (F_res_) corresponding to A_res_. It was demonstrated in the top graph of Figure 11 that the increase of the x value leads to an increase in the reflection losses maximum (as modulus) from −8.36 dB for x = 1 to −20 and −19.5 dB for x = 4 and x = 5 respectively.

Monotonic increase in the reflection coefficient corresponds to the resonance absorption processes. This could be due to an increase in the spinel phase in composite materials. It is well known that resonant frequency of absorption for spinels is observed in the MHz range (from several tens to several hundred MHz depending on the composition). Modification of the resonant frequency as a function of x is not linear and has complex behavior. Nevertheless, the range of F_res_ is not wide (min F_res_ = 91 MHz for x = 4 and max F_res_ = 121 MHz for x = 2). It means that absorption in investigated composites is characterized by resonant phenomena in spinel phase (for M-type hexaferrites F_res_~50-51 GHz). Non-linear shifts in F_res_ may be the result of intergranular interactions between two magnetic phases in composites.

## 4. Conclusions

Measurements of the MW characteristics for several typologies of nanoferrites [S_r0.3_Ba_0.4_Pb_0.3_Fe_12_O_19_/(CuFe_2_O_4_)_x_ (x = 2, 3, 4, and 5)] were performed using a coaxial method in the X-band (frequency range 8–12 GHz). It was observed that differences in values of the real part of permittivity and conductivity occur depending on the ferrites compositions. This non-linear behavior may be the result of different degrees of oxidation on spinel/hexaferrite grain boundaries with a high value of the activation energy of the intergranular potential barrier. As expected in the range of 8–12 GHz, the highest RL was found for x = 3 and 5 (less than −4.5 dB), meaning that the main losses are due to reflection rather than absorption. The strong coupling between phases was established by measurements performed using a coaxial method in the frequency range 1 MHz–1 GHz. In this frequency range there was a resonance behavior which is typical for spinels. It was demonstrated that the reflection losses maximum (as modulus) increases with an increasing x value from −9.1 dB (for x = 2) to −20–−19.5 dB (for x = 4 and x = 5 respectively). Monotonic increase in the reflection coefficient corresponds to the resonance absorption processes. This may be due to an increase in the spinel phase influence. Non-linear shifts in F_res_ may be the result of intergranular interaction between two magnetic phases in composites. Effective absorption of these composites opens broad perspectives for their exploitation in 4G-technology (information transfer) as well as for biomedical applications (for example, as magnetic nanoparticles for hyperthermic applications against cancer).

## Figures and Tables

**Figure 1 nanomaterials-09-00202-f001:**
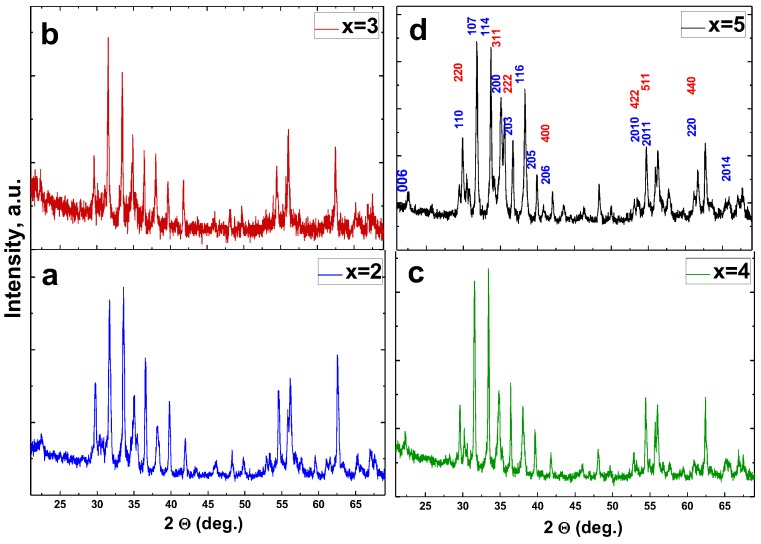
The X-ray diffraction powder patterns of the Sr_0.3_Ba_0.4_Pb_0.3_Fe_12_O_19_/(CuFe_2_O_4_)_x_ composite samples x = 2 (**a**), 3 (**b**), 4 (**c**), and 5 (**d**).

**Figure 2 nanomaterials-09-00202-f002:**
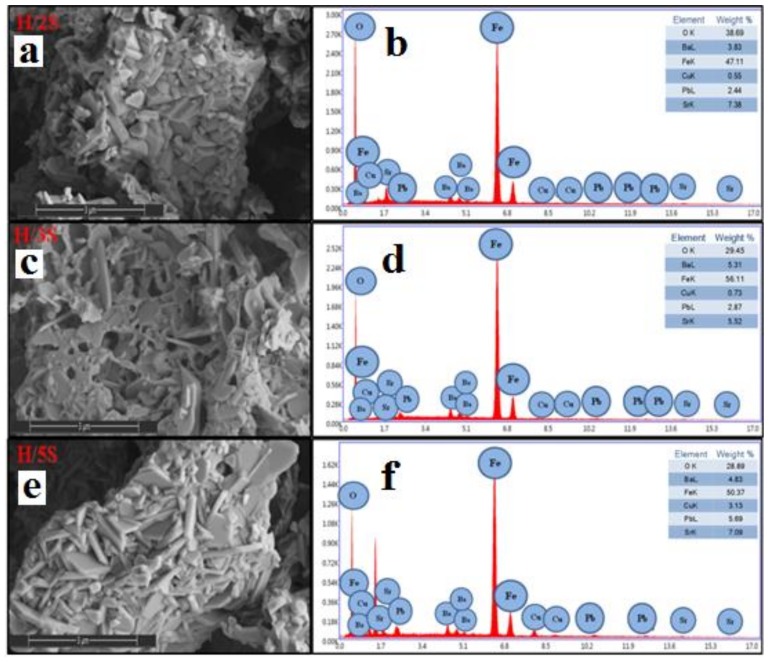
SEM images (a, c, e) and EDX spectra (b, d, f) of S_r0.3_Ba_0.4_Pb_0.3_Fe_12_O_19_/(CuFe_2_O_4_)_x_ composite samples x = 2 (**a**, **b**), 3 (**c**, **d**) and 5 (**e**, **f**).

**Figure 3 nanomaterials-09-00202-f003:**
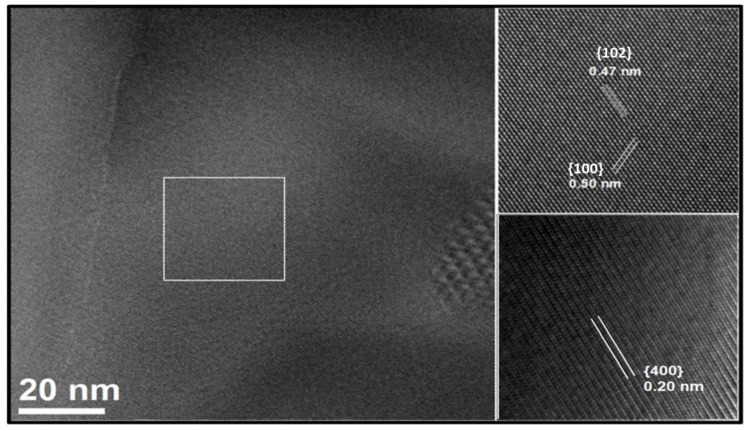
HRTEM images of S_r0.3_Ba_0.4_Pb_0.3_Fe_12_O_19_/(CuFe_2_O_4_)_x_ composite samples.

**Figure 4 nanomaterials-09-00202-f004:**
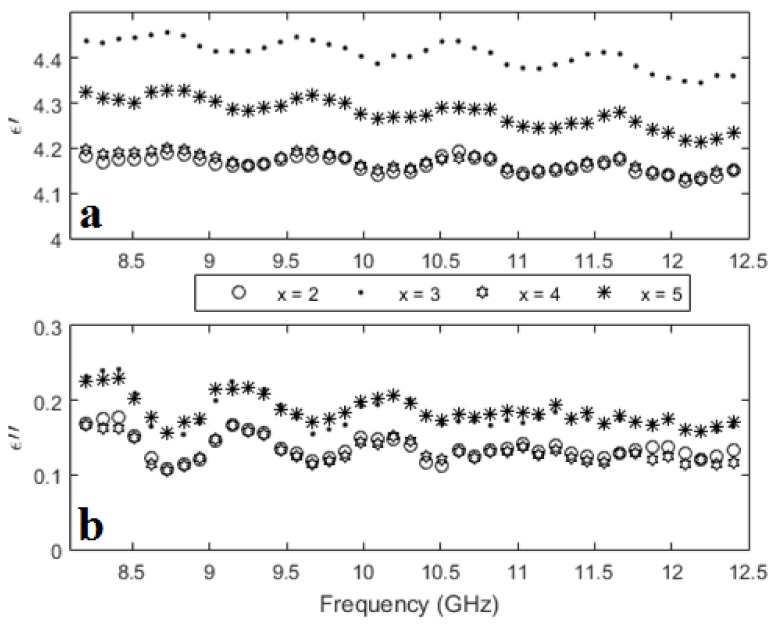
Frequency dependencies of permittivity: real (a) and imaginary (b) parts for S_r0.3_Ba_0.4_Pb_0.3_Fe_12_O_19_/(CuFe_2_O_4_)_x_ composite samples (x = 2, 3, 4 and 5).

**Figure 5 nanomaterials-09-00202-f005:**
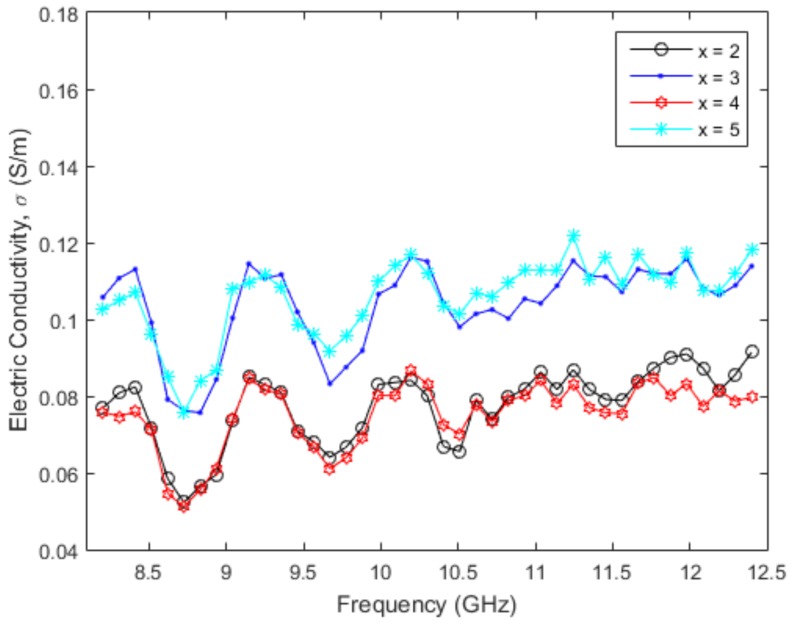
Frequency dependencies of electric conductivity for S_r0.3_Ba_0.4_Pb_0.3_Fe_12_O_19_/(CuFe_2_O_4_)_x_ composite samples (x = 2, 3, 4 and 5).

**Figure 6 nanomaterials-09-00202-f006:**
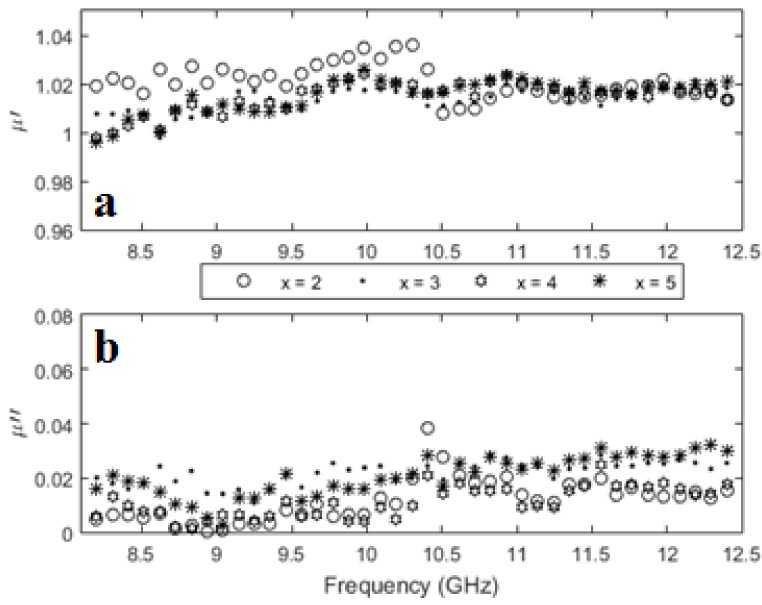
Frequency dependencies of permeability: real (**a**) and imaginary (**b**) parts for S_r0.3_Ba_0.4_Pb_0.3_Fe_12_O_19_/(CuFe_2_O_4_)_x_ composite samples (x = 2, 3, 4 and 5).

**Figure 7 nanomaterials-09-00202-f007:**
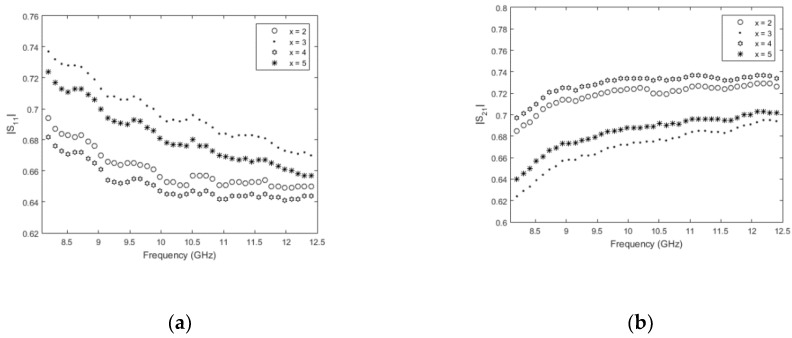
Frequency dependencies of (**a**) S11 and (**b**) S21 parts for S_r0.3_Ba_0.4_Pb_0.3_Fe_12_O_19_/(CuFe_2_O_4_)_x_ composite samples (x = 2, 3, 4 and 5).

**Figure 8 nanomaterials-09-00202-f008:**
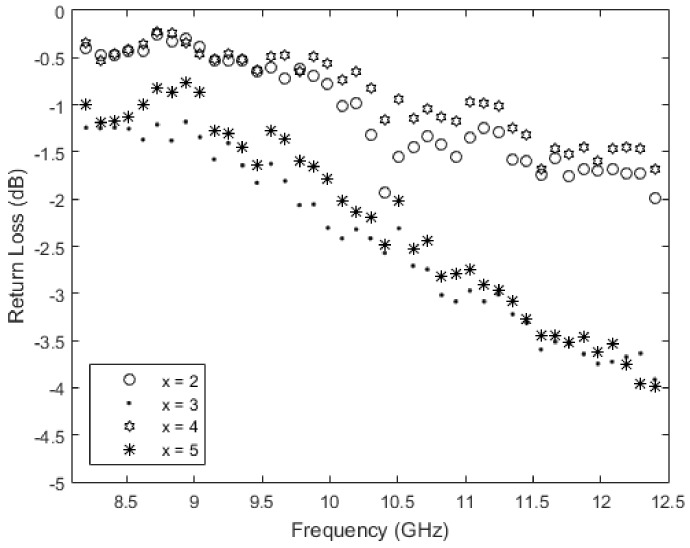
Frequency dependencies of reflection for S_r0.3_Ba_0.4_Pb_0.3_Fe_12_O_19_/(CuFe_2_O_4_)_x_ composite samples (x = 2, 3, 4 and 5).

**Figure 9 nanomaterials-09-00202-f009:**
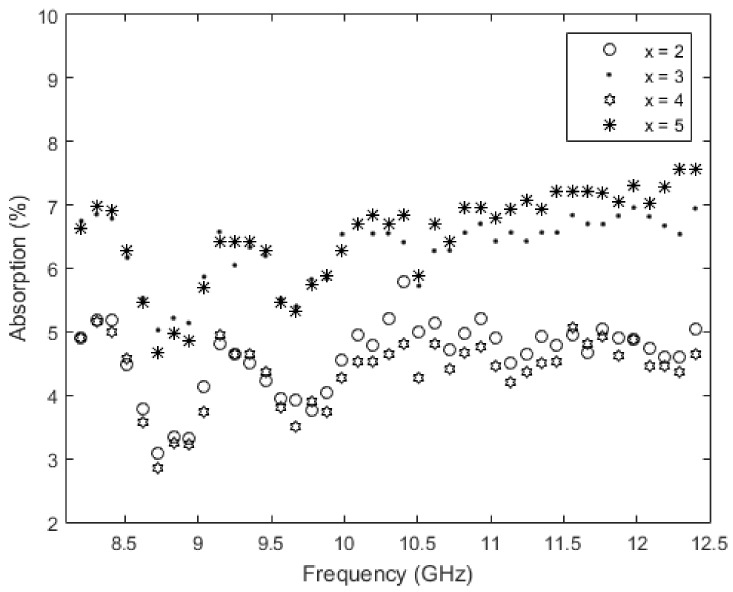
Frequency dependencies of absorption for S_r0.3_Ba_0.4_Pb_0.3_Fe_12_O_19_/(CuFe_2_O_4_)_x_ composite samples (x = 2, 3, 4 and 5).

**Figure 10 nanomaterials-09-00202-f010:**
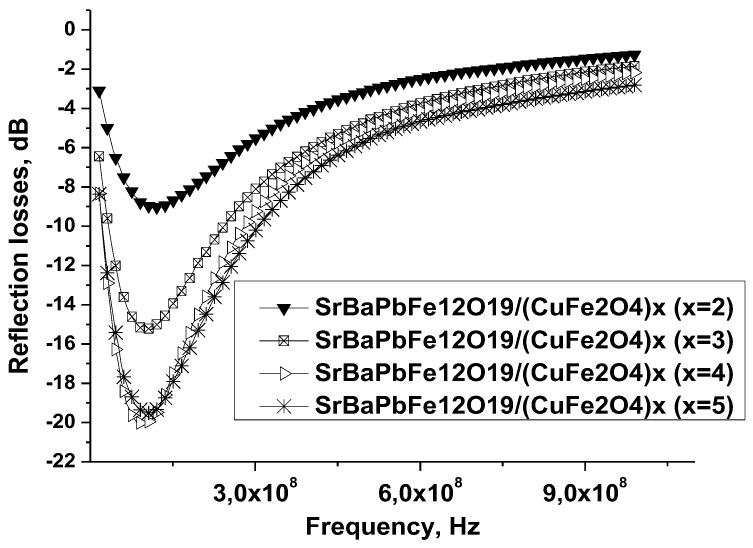
Frequency dependencies of the reflection losses (in dB) for S_r0.3_Ba_0.4_Pb_0.3_Fe_12_O_19_/(CuFe_2_O_4_)_x_ composite samples (x = 2, 3, 4 and 5).

**Figure 11 nanomaterials-09-00202-f011:**
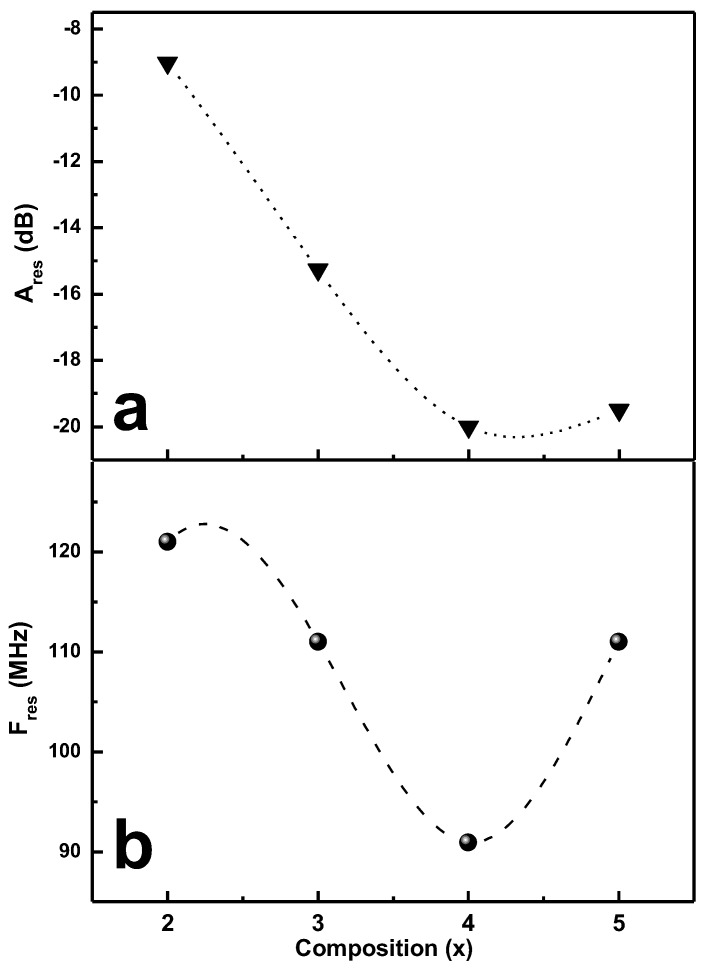
Composition dependencies of the resonant amplitude −A_res_ (a) and resonant frequency −F_res_ (b) for S_r0.3_Ba_0.4_Pb_0.3_Fe_12_O_19_/(CuFe_2_O_4_)_x_ composite samples (x = 2, 3, 4 and 5).

**Table 1 nanomaterials-09-00202-t001:** The features of the crystal structure (a and c unit cell parameters and D—average crystallite size) for each phase (Sr_0.3_Ba_0.4_Pb_0.3_Fe_12_O_19_ and CuFe_2_O_4_) of composites were obtained using XRD data. The data presented in this table were reused with permission from [30] (copyright 2018, Elsevier).

Composition	Sr_0.3_Ba_0.4_Pb_0.3_Fe_12_O_1_ hard magnetic phase (Å)		CuFe_2_O_4_ soft magnetic phase (Å)	D (nm)	
	a	c	a	Sr_0.3_Ba_0.4_Pb_0.3_Fe_12_O_19_	CuFe_2_O_4_
Sr_0.3_Ba_0.4_Pb_0.3_Fe_12_O_19_/(CuFe_2_O_4_)_2_ (1:2)	5.888	23.126	8.411	12.0	32.2
Sr_0.3_Ba_0.4_Pb_0.3_Fe_12_O_19_/(CuFe_2_O_4_)_3_ (1:3)	5.880	23.105	8.397	31.8	48.6
Sr_0.3_Ba_0.4_Pb_0.3_Fe_12_O_19_/(CuFe_2_O_4_)_4_ (1:4)	5.884	23.104	8.370	25.4	48.3
Sr_0.3_Ba_0.4_Pb_0.3_Fe_12_O_19_/(CuFe_2_O_4_)_5_ (1:5)	5.887	23.128	8.323	8.0	38.3

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
