# Peer review of "Correlation Between Composition and Electrodynamics Properties in Nanocomposites Based on Hard/Soft Ferrimagnetics with Strong Exchange Coupling"

_nanomaterials, 2019, doi:10.3390/nano9020202_

Reviewer 1 Report

The Authors improved the manuscript correctly – it can be now considered for publication.

Final minor suggestions to refine the text quality:

- … by using citric acid for and a chelation …

- … by widely adopted coaxial [31, 32] …

- Further investigations were carried out at T=300K in the range 8-12.5 GHz (split by punctuation) no critical changes …

- Figure 9 demonstrates the frequency dependence of absorption losses for the investigated composites investigated in the frequency range 8-12 GHz. It is clear that no any sufficient weakening is caused by electromagnetic absorption in this frequency range. It must be mentioned that this frequency range is a typical for region of absorption for ferrites with hexagonal structure. It means that the mixing of the hard magnetic and soft magnetic phases in composites leads to significant decrease in the microwave properties of the hexagonal ferrites (hard magnets). Figure 10 demonstrates the frequency dependence of absorption losses. These measurements were performed measured by using coaxial method in the frequency range 1 MHz – 1 GHz. This frequency region was chosen due to the possibility to observeation of resonance reflection (NFMR) for soft magnetic phase (spinel). It must can be noted the presence of resonance (significant weakening of the reflected radiation) in this frequency range. In this frequency range, a resonance as typical of spinels was observed. Such behavior This is due to a strong coupling ...

- meaningh

Author Response

Response for #1_Reviewer’s comments

on paper “Correlation between composition and electrodynamics properties in nanocomposites based on hard/soft ferrimagnetics

with strong exchange coupling” (nanomaterials-438294

by authors M.A. Almessiere, A.V. Trukhanov, Y. Slimani, K.Y. You, S.V. Trukhanov, E.L. Trukhanova, M. Zdorovets, F. Esa, A. Sadaqat, K. Chaudhary, A. Baykal

Comments and Suggestions for Authors

The Authors improved the manuscript correctly – it can be now considered for publication.

Final minor suggestions to refine the text quality:

- … by using citric acid for and a chelation …

Response to comment: Dear Reviewer, many thanks for your comment. We did this.

- … by widely adopted coaxial [31, 32] …]\

Response to comment: Dear Reviewer, many thanks for your comment. We did this.

- Further investigations were carried out at T=300K in the range 8-12.5 GHz (split by punctuation)no critical changes …

Response to comment: Dear Reviewer, many thanks for your comment. We did this.

- Figure 9 demonstrates the frequency dependence of absorption losses for the investigatedcomposites investigated in the frequency range 8-12 GHz. It is clear that no any sufficient weakening is caused by electromagnetic absorption in this frequency range. It must be mentioned that this frequency range is a typical for region of absorption for ferrites with hexagonal structure. It means that the mixing of the hard magnetic and soft magnetic phases in composites leads to significant decrease in the microwave properties of the hexagonal ferrites (hard magnets). Figure 10 demonstrates the frequency dependence of absorption losses. These measurements were performed measured by using coaxial method in the frequency range 1 MHz – 1 GHz. This frequency region was chosen due to the possibility to observeation of resonance reflection (NFMR) for soft magnetic phase (spinel). It must can be noted the presence of resonance (significant weakening of the reflected radiation) in this frequency range. In this frequency range, a resonance as typical of spinels was observedSuch behavior This is due to a strong coupling ...

Response to comment: Dear Reviewer, many thanks for your comment. We did this.

- meaningh

Response to comment: Dear Reviewer, many thanks for your comment. We did this

Dear Reviewer,

We did all corrections in accordance with your comments. We hope that this version of the paper is suitable for publication in Nanomatarials.

Reviewer 2 Report

In order to read and review the manuscript more easily the authors shall 1) accept all changes in the manuscript and not showing the track changes and 2) use the MPDPI template and include the text. 

In the abstract and in the result chapter it is claimed that the XRD analysis shows that the material has strong exchange coupling and in the result chapter it is referred to ref. 18 where the magnetic analysis are shown. Add some text in the manuscript about the exchange coupling analysis and result.

Please, go through the references (for instance 18 and 19). It shall be Materials.

Author Response

Response for #2_Reviewer’s comments

on paper “Correlation between composition and electrodynamics properties in nanocomposites based on hard/soft ferrimagnetics

with strong exchange coupling” (nanomaterials-438294)

by authors M.A. Almessiere, A.V. Trukhanov, Y. Slimani, K.Y. You, S.V. Trukhanov, E.L. Trukhanova, M. Zdorovets, F. Esa, A. Sadaqat, K. Chaudhary, A. Baykal

Comments and Suggestions for Authors

Comment 1: In order to read and review the manuscript more easily the authors shall 1) accept all changes in the manuscript and not showing the track changes and 2) use the MPDPI template and include the text. 

Response to comment: Dear Reviewer, many thanks for your comment. We did this. We accepted all changes and didn’t show any track changes. We used MDPI template.

Comment 2: In the abstract and in the result chapter it is claimed that the XRD analysis shows that the material has strong exchange coupling and in the result chapter it is referred to ref. 18 where the magnetic analysis are shown. Add some text in the manuscript about the exchange coupling analysis and result.

Response to comment: Dear Reviewer, many thanks for your comment. We did this. 

Comment 3: Please, go through the references (for instance 18 and 19). It shall be Materials.

Response to comment: Dear Reviewer, many thanks for your comment. We did this. 

Dear Reviewer,

We did all corrections in accordance with your comments. We hope that this version of the paper is suitable for publication in Nanomatarials

Round  2

Reviewer 2 Report

The authors have improved the manuscript. I have only minor comments (see below).

Abstract: line 33. write nanosized ferrites.

line 90: write real and imaginary parts of permittivity and permeability

line 147: write some text about the analysis in ref 18 and how it was concluded that the material have a strong exchange coupling.

Author Response

Dear Reviewer,

Many thanks for your help and kindness for us.

We made revision in accordance with your Comments and Suggestions.

Please see it in revised version.

Comment #1 The authors have improved the manuscript. I have only minor comments (see below).

 Response - Many thanks. We did this.

Comment #2Abstract: line 33. write nanosized ferrites.

 Response - We did this.

Comment #3 line 90: write real and imaginary parts of permittivity and permeability

 Response - We did this.

Comment #4 lline 147: write some text about the analysis in ref 18 and how it was concluded that the material have a strong exchange coupling.

Response - We did this.

This manuscript is a resubmission of an earlier submission. The following is a list of the peer review reports and author responses from that submission.

Round  1

Reviewer 1 Report

The paper reports a microwave characterization of ferrite-based nanomaterials, focusing on the influence of the different magnetic phases in the composites. Both the manufacturing procedure and the qualitative and quantitative results may be of interest for the skilled reader; nevertheless, in my opinion, the manuscript needs of extending editing in order to be published.

Most of all, the quality of presentation is very poor and a careful revision of the language is mandatory - only few examples:

- adjust the symbols used for Angström (rows 50-51) and Celsius degrees (rows 74,78);

- row 62: correct “method that lead(s)”;

-  adjust syntax at row 80: “…obtain initial powders. Which were…”;

- row 83: correct “…were analyzed (by) SEM…”;

- row 94: correct “…coaxial line impedance (is) determined…”;

- row 116: rearrange “It was observed that in all samples”;

- row 145: “(at. %)” ??

- row 159: rearrange “For this investigation can be used the long-line method”

- rows 168-169: “…were placed in co-axial line (Z=50 Ohm) and investigated electrical parameters.”;

- row 178: correct “…to absence (of) any electrical…”;

- rows 180-181: rearrange “For real part of composites observed dispersion in values for samples with x=2, 4 and other.”;

- rows 201-202: correct “…to absence (of) any magnetic…”;

Etc. etc……………

The “wide frequency range” claimed at row 86 is questionable, being more rightly associate to 2-18 GHz ; the range analyzed (8.2-12.4 GHz) is more properly defined as X-band. At the end of the work the analysis is suddenly shifted to a different range; that represents a bit of a mess, also because no mention about the different method of characterization is provided.

Author Response

Response for Reviewer’s comments on paper “Correlation between composition and electrodynamics properties in nanocomposites based on hard/soft ferrimagnetics with strong exchange coupling” by authors M.A. Almessiere, A.V. Trukhanov, Y. Slimani, K.Y. You, S.V. Trukhanov, F. Esa, A. Sadaqat, K. Chaudhary, A. Baykal 

 Reviewer 1 Comments and Suggestions for Authors 

The paper reports a microwave characterization of ferrite-based nanomaterials, focusing on the influence of the different magnetic phases in the composites. Both the manufacturing procedure and the qualitative and quantitative results may be of interest for the skilled reader; nevertheless, in my opinion, the manuscript needs of extending editing in order to be published. Most of all, the quality of presentation is very poor and a careful revision of the language is mandatory - only few examples: - adjust the symbols used for Angström (rows 50-51) and Celsius degrees (rows 74,78); - row 62: correct “method that lead(s)”; -  adjust syntax at row 80: “…obtain initial powders. Which were…”; - row 83: correct “…were analyzed (by) SEM…”; - row 94: correct “…coaxial line impedance (is) determined…”; - row 116: rearrange “It was observed that in all samples”; - row 145: “(at. %)” ?? - row 159: rearrange “For this investigation can be used the long-line method” - rows 168-169: “…were placed in co-axial line (Z=50 Ohm) and investigated electrical parameters.”; - row 178: correct “…to absence (of) any electrical…”; - rows 180-181: rearrange “For real part of composites observed dispersion in values for samples with x=2, 4 and other.”; - rows 201-202: correct “…to absence (of) any magnetic…”; Etc. etc……………

Response: Dear Reviewer many thanks for this comment. Of course in previous version there were a lot of grammatical mistakes and typos. We made strict revision and improved it. We hope that now paper seems better.

The “wide frequency range” claimed at row 86 is questionable, being more rightly associate to 2-18 GHz ; the range analyzed (8.2-12.4 GHz) is more properly defined as X-band. At the end of the work the analysis is suddenly shifted to a different range; that represents a bit of a mess, also because no mention about the different method of characterization is provided.

Response: Dear Reviewer many thanks for this comment. We really performed all measurements of electrodynamic characteristics for X-band due to our proposal that strong coupling between soft and hard magnetic phases may change the properties in this frequency range. But after measurements and careful result evaluation we decided to make reflection investigations for 100 MHz – 1 GHz range. And we want to demonstrate that soft magnetic phase has no critical influence on hard phase characteristics (in X-band). And hard phase changes amplitude-frequency characteristics.

Sincerely your’s,

author’s team.

Reviewer 2 Report

All equations and modeling methods are known.

No nano-effect is related to the experimental data.

The wavy nature (Figs. 4-6) of the experimental data is questionable.

This Ms is weakly interpreted.

The Ms is written in poor English and contains much typos.

Overall, I do not recommend publication of this Ms because it does not add any substantial physical content to the existing literature. That is not applied physics at the level is expected. This Ms appears to contain a presentation of data without the in-depth analysis expected. Hence it is not sufficient to report outcomes of experiments without drawing out new physics in the interpretation of the results. In this Ms, there are descriptions about the Figures, but there is little said about their physical significance and what new understanding can be gained from them.

Author Response

Response for Reviewer’s comments

on paper “Correlation between composition and electrodynamics properties in nanocomposites based on hard/soft ferrimagnetics

with strong exchange coupling”

by authors M.A. Almessiere, A.V. Trukhanov, Y. Slimani, K.Y. You, S.V. Trukhanov, F. Esa, A. Sadaqat, K. Chaudhary, A. Baykal

Reviewer 2

Comments and Suggestions for Authors

1. All equations and modeling methods are known.

Response: Dear Reviewer many thanks for this comment. All equations and modeling methods are known – we used standard methods.

2. No nano-effect is related to the experimental data.

Response: Dear Reviewer many thanks for this comment. Yes, there is no “nano-effect” in investigated samples. We described results for nanosized composite materials. In further investigations we will compare properties for microsized and nanosized composites. And we will evaluate influence of the grain size. Many thanks for this idea.

3. The wavy nature (Figs. 4-6) of the experimental data is questionable.

Response: Dear Reviewer many thanks for this comment. But we can’t clearly understand what is questionable on your opinion? We think that all data is reliable. But thank you again.

4. This Ms is weakly interpreted.

Response: Dear Reviewer many thanks for this comment. Am I right that Ms is magnetization saturation? If yes, I should note that we didn’t discuss in present paper magnetic properties of the composites. Magnetic properties in static (field dependences of magnetization) were not the aim of this paper. It will be material for a further paper.

5. The Ms is written in poor English and contains much typos.

Response: Dear Reviewer many thanks for this comment. Of course in previous version there were a lot of grammatical mistakes and typos. We made strict revision and improved it. We hope that now paper seems better.

6. Overall, I do not recommend publication of this Ms because it does not add any substantial physical content to the existing literature. That is not applied physics at the level is expected. This Ms appears to contain a presentation of data without the in-depth analysis expected. Hence it is not sufficient to report outcomes of experiments without drawing out new physics in the interpretation of the results. In this Ms, there are descriptions about the Figures, but there is little said about their physical significance and what new understanding can be gained from them.

Response: Dear Reviewer many thanks for this comment. Of course this is your own opinion about scientific and practical significance. But we believe if you give us a chance and evaluate revised paper with “fresh eyes” we can have a good publication in a high impact journal. We hope on your understanding and your help. We think that in future this paper can be cited in many journals.

Sincerely your’s,

author’s team.

Round  2

Reviewer 1 Report

Authors tried to improve the manuscript, but - in my opinion - the quality of the presentation is still low in the revised version; an extensive editing and revisiting process is strongly suggested in order to make the content of the work worth of publication in a scientific journal.

Author Response

Response for Reviewer’s comments

on paper “Correlation between composition and electrodynamics properties in nanocomposites based on hard/soft ferrimagnetics

with strong exchange coupling”

by authors M.A. Almessiere, A.V. Trukhanov, Y. Slimani, K.Y. You, S.V. Trukhanov, F. Esa, A. Sadaqat, K. Chaudhary, A. Bayka

Reviewer 1

Comments and Suggestions for Authors

Authors tried to improve the manuscript, but - in my opinion - the quality of the presentation is still low in the revised version; an extensive editing and revisiting process is strongly suggested in order to make the content of the work worth of publication in a scientific journal.

Response: Dear Reviewer many thanks for this comment. We try to improve our manuscript again. Please, evaluate it again and let me know is it appropriate for publication in present form. We are strongly grateful you for your help. We hope on your understanding.

Sincerely your’s,

author’s team.

Reviewer 2 Report

1. All equations and modeling methods are known: The Authors confirmed!

2. No nano-effect is related to the experimental data: The Authors confirmed!

3. The wavy nature (Figs. 4-6) of the experimental data is questionable: Most of the experimental methods in this range of frequencies give smooth variations of the experimental data. Here, strong (wavy) fluctuations are observed which make me believe that the Authors do not control their protocol.

4. This Manuscript is weakly interpreted. Nothing new is modified in the revision!

5. There are still much typos and poor English.

6. Overall, I do not recommend publication of this Ms because it does not add any substantial physical content to the existing literature. That is not applied physics at the level is expected. This Ms appears to contain a presentation of data without the in-depth analysis expected. Hence it is not sufficient to report outcomes of experiments without drawing out new physics in the interpretation of the results. In this Ms, there are descriptions about the Figures, but there is little said about their physical significance and what new understanding can be gained from them.

Author Response

Response for Reviewer’s comments

on paper “Correlation between composition and electrodynamics properties in nanocomposites based on hard/soft ferrimagnetics

with strong exchange coupling”

by authors M.A. Almessiere, A.V. Trukhanov, Y. Slimani, K.Y. You, S.V. Trukhanov, F. Esa, A. Sadaqat, K. Chaudhary, A. Bayka

Reviewer 2

Comments and Suggestions for Authors

1. All equations and modeling methods are known: The Authors confirmed!

Response: Dear Reviewer many thanks for this comment. Yes, we did it.

2. No nano-effect is related to the experimental data: The Authors confirmed!

Response: Dear Reviewer many thanks for this comment. Yes, we did it.

3. The wavy nature (Figs. 4-6) of the experimental data is questionable: Most of the experimental methods in this range of frequencies give smooth variations of the experimental data. Here, strong (wavy) fluctuations are observed which make me believe that the Authors do not control their protocol.

Response: Dear Reviewer many thanks for this comment. I want to disagree with the statement. We controlled the measurements. The average values are given (measurements were automatic and fixed using software for 10 points at the frequency). Of course, there is a possibility an error; however, according to the hardware error data, they are insignificant.

4. This Manuscript is weakly interpreted. Nothing new is modified in the revision!

Response: Dear Reviewer many thanks for this comment. In modified  version we improve errors and language edition.

5. There are still much typos and poor English.

Response: Dear Reviewer many thanks for this comment. We try to improve our manuscript again with help of native speaker.

6. Overall, I do not recommend publication of this Ms because it does not add any substantial physical content to the existing literature. That is not applied physics at the level is expected. This Ms appears to contain a presentation of data without the in-depth analysis expected. Hence it is not sufficient to report outcomes of experiments without drawing out new physics in the interpretation of the results. In this Ms, there are descriptions about the Figures, but there is little said about their physical significance and what new understanding can be gained from them.

Response: Dear Reviewer many thanks for this comment. We are strongly grateful you for your help and opinion. We hope on your understanding.

Sincerely your’s,

author’s team.

Round  3

Reviewer 1 Report

Authors are deserving for their efforts. As only aid, I can encourage the publication of the manuscript by giving a full correction of the text hereafter.

About the topic of nano-structured and advanced materials for MW absorption, in the block of references [2-5] one should refer also to the following recent and well cited publications:

Carbon 144 (2019) 63-71

Journal of Building Engineering 18 (2018) 33-9

IEEE Transaction on Microwave Theory and Techniques 65 (2017) 2801-9

Carbon 77 (2014) 756-74

Bare essential corrections for eventual publication (add green, delete red):

ABS

…were fabricated by a one-pot sol–gel combustion…

1 INTRO

…as a result leads to an improvement of the functional properties. For example it can leads to modification…

The main aim is in the determination of to determine the correlation…

In case when As strong exchange-coupling is occurreds between two soft and hard magnetic phases, we can observe an intensification of the microwave absorption can be observed

2 EXP

Weight ratio between citric acid and ions nitrites was 1.5:1 respectively. After that, the mixture was…

We used citric acid for pH correction by using citric acid and chelation were used.

This first stage lets us allows to obtain initial powders.

we calculated …the reflection coefficients were calculated. For this, we used the calculation formula from by referring to the theory of propagation...

Where…Where…: use lowercase letter

For reflection losses, we used the next formula were used:

where |?̇| the modulus of the reflection

3 RES

…proved the coexistence of only two main…

…as function of frequency are extensively measured. For measurements as a rule by widely adopted coaxial [19, 20] and waveguides methods [21-23] are widely used.

In this case, waves rapidly decay for frequencies below the measurement range because their frequencies approach to the critical frequency of the waveguide. Under this critical frequency, waves in the waveguide cannot propagate. It means that Due to issues related to measurement dynamical range, the waveguides method is appropriate for narrow frequency ranges only, and limited by the size of the sample.

… it is required to have different measuring sections of the waveguide with different cross-sections and to prepare appropriate sizes of the samples of suitable size.

In the investigated range no observed critical changes in permittivity due to absence of any electrical losses in composites were observed in 8-12.5 GHz.

…were calculated using the following equations:

Where…: use lowercase letter

Figure 5 demonstrates the frequency dependences of the electrical conductivity. The values measured for the of conductivity is are typical for highly doped semiconductors or composites in which realized where hopping conduction mechanism occurs. It was observed the same The revealed blurred peaks are in agreement with the measurement of permittivity in the same MW range.

Figure 6 demonstrates the frequency dependences of permeability (real and imaginary parts). Further investigations also were carried out at T=300K in the range 8-12.5 GHz. In the investigated range : no observed critical changes in permeability due to absence of any magnetic losses in composites were observed in this region.

…using the Nicholson-Ross-Weer method (NRW) [25] for which S by measuring the as-called scattering parameters of the coaxial line segment S11 and S21 were measured.

Figure 7 a and b demonstrate the frequency dependences of the S11-S21 parameters.

The maximal values of RL for x=3 and 5 are less than -4.5 dB. It means , thus the main losses are due to reflection not rather than absorption (if RL<-10 dB main mechanism is reflection)

Using transmission TL and reflection RL coefficients we calculated the absorption coefficient ???? was calculated as follows:…

Figures 9 and demonstrates the frequency dependence of absorption. Figure 10 demonstrates the frequency dependences of the absorption and reflection losses, respectively.

In this frequency range, a resonance was observed which is typical for of spinels was observed. Figure 8 demonstrates the concentration dependences of the maximum value of reflection or resonant amplitude (Ares) and of the resonant frequency (Fres) which corresponding to Ares.

It is well known that resonant frequency of absorption for spinels is observed in the MHz range (from several tens to several hundred MHz in dependence of composition). Modification of the resonant frequency as a function of x is not linear and has complex behavior. But , nevertheless the frequency range of for Fres is not wide …

4 CON

Measurements of MW characteristics of several typologies of nanoferrites

[ Sr0.3Ba0.4Pb0.3Fe12O19/(CuFe2O4)x - (x = 2, 3, 4, and 5) ] were performed using coaxial method in the X-band (frequency range 8-12 GHz). It was observed that differences in values of real part of permittivity and conductivity occur depending on the for samples with x=2, 4 and other ferrites compositions.

Maximal values of RL aAs expected in the range 8-12 GHz, the highest RL was determined for x=3 and 5 (less than -4.5 dB). It means , meaning that the mainly these losses are due to reflection not rather than absorption (if RL<-10 dB main mechanism is reflection). For analysis of tThe strong coupling between phases was established by measurements were performed using coaxial method in the frequency range 1 MHz – 1 GHz. In this frequency range there was observed a resonance behavior which is typical for spinels.

Effective absorption of these composites opens broad perspectives for its application  their exploitation in 4G-technology (information transfer) and , as well as for biomedical applications (for example, as magnetic nanoparticles for hyperthermic applications against cancer).

Reviewer 2 Report

I do not recommend publication of this Ms because it does not add any substantial physical content to the existing literature. That is not applied physics at the level is expected. This Ms appears to contain a presentation of data without the in-depth analysis expected. Hence it is not sufficient to report outcomes of experiments without drawing out new physics in the interpretation of the results. In this Ms, there are descriptions about the Figures, but there is little said about their physical significance and what new understanding can be gained from them.